# dTOURS: Dense-region tagging for outbreak detection using ratio statistics

**Lukas Wagner[1], Richa Agarwala[2]***

**1** Emeritus, National Center for Biotechnology Information, Bethesda, Maryland, United States of America,
**2** National Center for Biotechnology Information, Bethesda, Maryland, United States of America

\* agarwala@ncbi.nlm.nih.gov

## Abstract

Surveillance for food safety in the United States of America is a collaborative effort among public health agencies with additional partners worldwide contributing sequence data. Assemblies in GenBank and sequence reads in the Sequence Read Archive for surveilled species are received, rapidly analyzed, and results published publicly by an automated pathogen detection pipeline at the National Center for Biotechnology Information. The pipeline detects close isolates with a recent common ancestor by finding single nucleotide polymorphisms (SNPs) in genomes for pairs of isolates. Very few vertically transmitted SNPs are expected between a pair of close isolates; any genomic region with many SNPs compared to the number of SNPs in the rest of the genome is indicative of a horizontally transferred region that needs to be excluded for counting vertically transmitted SNPs. We developed dTOURS that adapted the ratio statistic for finding outliers to the problem of finding regions of high SNP density in a pair of genomes where isolates typically have fragmented genome assemblies. Simulations for deciding the dTOURS parameter are presented. We illustrate correctness of dTOURS using five published outbreaks, one each for five bacterial species that cause many foodborne outbreaks or lead to a high mortality rate. Comparison to Gubbins shows that while both Gubbins and dTOURS use the ratio statistic, the implementation in dTOURS is more robust for finding close isolates in outbreak analysis. Comparison with the method used by the Food and Drug Administration shows that their method is simple and fast but not sensitive.

## Introduction

A collaborative effort between the Food and Drug Administration (FDA), U.S. Centers for Disease Control and Prevention (CDC), and the National Center for Biotechnology Information (NCBI) to ensure food safety using whole genome sequencing started in 2013 with a pilot to monitor four bacterial foodborne pathogens. Sequencing and enforcement efforts are carried out by multiple networks [1,2], including PulseNet at

**Data availability statement:** Software is available at https://doi.org/10.5281/zeno-do.14532372 Sequences used for analysis are publicly available at NCBI. Additional data is provided as Supplementary Information.

**Funding:** This work was supported by the National Center for Biotechnology Information of the National Library of Medicine (NLM), National Institutes of Health. There was no additional external funding received for this study. The contents of this publication are solely the responsibility of the authors and do not necessarily represent the official views of the National Center of Biotechnology Information

**Competing interests:** The authors have declared that no competing interests exist.

CDC [3] and GenomeTrakr at FDA [4]. The NCBI Pathogen Detection Pipeline [5] analyzes the sequences submitted to the pathogen detection project (PDP) by continuously pulling assemblies from GenBank® and reads sequenced using Illumina from the Sequence Read Archive [6] for species it is tracking. As of November 2024, PDP was tracking 756 bacterial and fungal species in 99 organism groups, and it had analyzed over 2 million isolates. As of September 2024, GenomeTrakr had taken 1,332 actions intended to protect consumers from foodborne illness [4].

An important component of the analysis pipeline for finding the source of the common ancestor causing an outbreak is finding single nucleotide polymorphisms (SNPs) between isolates accumulated during vertical transmission. Recent vertical transmission of the chromosome from an ancestor to progenies in microbes introduce a few SNPs. On the other hand, horizontally transferred regions, such as plasmids and phages, typically have a higher density of SNPs (HDR) and can significantly impact pairs considered to be close [7,8]. Most pipelines for detecting outbreaks causing foodborne illness account for such regions in two main ways [9–18]. They either (i) limit the analysis to only the "core" genome expected to be present in all isolates, thereby minimizing the chance of including potential HDRs, or (ii) they incorporate a method to filter HDRs in the "whole" genome analysis. The whole genome analysis is computationally more expensive than the core genome analysis but has higher resolution [9,10]. Both the core genome and whole genome comparative analyses typically use a gene-by-gene approach, also known as multi locus sequence typing [11–13], or single nucleotide polymorphisms (SNPs) [14–18]. This manuscript presents a method called dTOURS (pronounced *dee-tours*) for detecting HDRs in the whole genome SNP analysis pipeline used by PDP.

Finding HDRs in the whole genome analysis is typically done in two main ways: (i) by relying on a reference set of known mobile genetic elements to mask regions on the genome before calling SNPs so no SNPs are called in the masked regions and (ii) by using a SNP density measure for HDR detection after calling SNPs; regions with a high density compared to the overall density of SNPs between pairs of genomes are considered as likely to be horizontally transferred and not inherited vertically from a common ancestor. For example, the CDC's Lyve-set SNP calling pipeline [15] uses a reference genome for calling SNPs. The genome alignment to PHAST database [19] is used for masking phage regions found in the reference genome. The pipeline also excludes SNPs that are nearby each other; typically, atmost 5 bp for *Salmonella*. FDA's CFSAN SNP pipeline [16] removes SNPs when more than 3 SNPs are found in a 1000 bp region of the genome. Gubbins [20] detects HDRs using the well-known ratio statistic measure for finding outliers [21]. It is available as a pipeline that uses SKA aligner [22] and as a filter after SNPs have been called in pipelines, such as, in SNVPhyl [17] and Snippy [18].

Gubbins is used for building phylogenies and removing HDRs in close as well as diverged isolates. It uses windows of various pre-selected sizes to scan for density variations. Motivated by the application of Gubbins in [23], we developed dTOURS that adapted the ratio statistic to our scenario of many fragmented genomes where we are only interested in finding close isolates. dTOURS is fully automated, does not

rely on any reference set or pre-selected window sizes, and has a single parameter for the log likelihood ratio used as the threshold for the binary decision of tagging a genomic region as dense or not. The parameter is empirically determined *once* using the expected length of the genome and the maximum number of vertically transmitted SNPs that can be present between a pair of isolates considered as part of an outbreak.

Here we present the algorithm design of dTOURS, simulations used to determine the parameter for dTOURS, and comparison of SNPs after dTOURS filtering for five published outbreaks: one each for *Campylobacter jejuni* [24], *Legionella pneumophila* [25], *Listeria monocytogenes* [26], *Escherichia coli* [27], and *Salmonella Typhimurium* [28]. We also present comparison to Gubbins using the same five outbreaks. While the SNP counts after dTOURS and Gubbins filtering are similar, we show that dTOURS is more robust than Gubbins for detecting close isolates in two main ways. First, dTOURS is not affected by the order of contigs in an assembly. This is particularly important in a production pipeline for regression testing and for providing the same set of SNPs for essentially the same assemblies as contig order does not have a meaning in fragmented assemblies. Second, dTOURS is more precise as it looks at all possible regions whereas Gubbins cannot afford to do the same as Gubbins is designed for both close and distant isolates. Comparison using simulations show that FDA method is fast but not sensitive. The software for dTOURS is available at https://doi.org/10.5281/zenodo.14532372

## Materials and methods

### dTOURS algorithm

dTOURS implements the spatial scan statistic using the Bernoulli model as proposed by Kulldorff for identifying dense clusters of arbitrary sizes in spatially distributed datasets [21]. The statistic allows for variable window size and performs a likelihood ratio test to compare the values in the window to the expected value under the null hypothesis of a random spatial distribution. The likelihood computation for our application of SNPs between a pair of genomes P and region R has following variables:

n denotes the total number of SNPs in region R,
g denotes the length of region R,
N denotes the total number of SNPs for pair P, and
G denotes the total length of the alignments for pair P.

With above variables, the null hypothesis log likelihood is computed as

$$LL^{NULL} = N * \log(N/G) + (G-N) * \log(1 - N/G)$$

Log likelihood for the SNP counts in region R and remaining SNPs in the rest of the genome alignments is computed as

$$LL^{REGION} = n * \log(n/g) + (g-n) * \log(1 - n/g) +$$

$$(N-n) * \log((N-n)/(G-g)) + ((G-g)-(N-n)) * \log(1 - ((N-n)/(G-g)))$$

The ratio statistic RS for the region is then computed as

$$RS = LL^{REGION} - LL^{NULL}$$

We implemented above iteratively where regions that achieve the maximum score [max(RS)] are filtered as dense regions if max(RS) is above the specified threshold. Fig 1 presents an overview for the approach. All regions delimited by every pair of SNPs for P are eligible for finding max(RS) in the first iteration; later iterations remove from consideration any regions that overlap dense regions filtered so far. After each iteration the count of SNPs in dense regions removed and

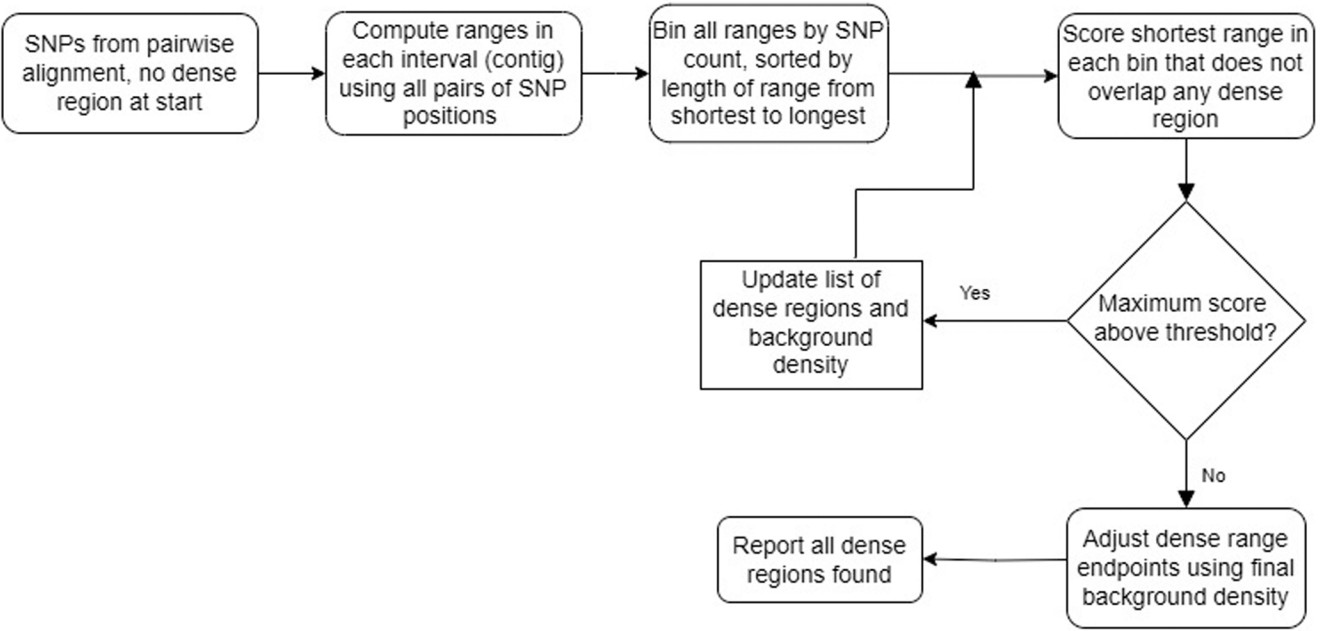

**Fig 1. Flowchart for dTOURS algorithm.** Input is a set of SNPs on a pair of genomes and output is HDRs.

the total lengths of those region are subtracted from N and G values, respectively, to reflect the new background rate. The iterations are terminated when max(RS) is below the threshold. The main optimization in the implementation was maintaining sorted lists for all regions not marked as dense and not intersecting any dense region found so far. Each list had regions with the same SNP count and sorting was done from the shortest to the longest region. With these lists, in each iteration, only the regions with the shortest length for each SNP count needs to be checked for finding the maximum RS value; for the same SNP count, shorter region has higher RS than a longer region.

After completing all iterations, count of SNPs not removed in dense regions and the length of the genome after removing total length of dense regions provides a better background estimate B of true rate of vertical transmission than the rate available in each iteration. Therefore, after completing all iterations, for every region R found as dense and estimate B, we expand R using SNPs near it and without intersecting another dense region if doing so gives a higher RS than with R.

### dTOURS implementation for fragmented genomes

For each pair, the genome assembly with higher N50 is considered as subject for alignment by BLAST [29] while the one with smaller N50 is the query. Consecutive aligned blocks on the same subject contig are concatenated together to produce aligned sequences and positions for SNPs found by BLAST adjusted accordingly. Only pairs where at least 80% of both genome assemblies are present in aligned sequences are considered as potentially close isolates and considered for dTOURS filtering.

Regions delimited by SNPs respect subject contig boundaries and are restricted to within the same contig. With this restriction, the order in which contigs appear in an assembly does not affect the results. Fig 2 shows an example of SNPs called between genome assemblies GCA_011671145.1 and GCA_010507495.1. The example provides a visual view for filtered regions and shows calculations done in each iteration. Total of 1446 SNPs were found on four contigs in 4,901,276 bp of aligned regions. All 916, 484, and 45 SNPs found on aligned blocks of contigs DAARVY010000032.1, DAARVY010000034.1, and DAARVY010000047.1, respectively, were filtered. The RS values in the three iterations were 3609.52, 2289.05, and

**Fig 2. SNPs between GCA_011671145.1 and GCA_010507495.1 before filtering.** SNPs are color coded by contig. Coordinates on the X-axis are by concatenating aligned blocks of contigs in the order in which they appear in the subject genome GCA_010507495.1.

289.53. These had (n,g,N,G) values as (916,34356,1446,4901276) with 916 SNPs found on DAARVY010000032.1 in 34,356 bp of consecutively aligned regions, (484,31374,530,4866920) in the second iteration with 916 SNPs and 34,356 bp removed from the background and 484 SNPs found on DAARVY010000034.1 in 31,374 bp of consecutively aligned regions, and (45,7000,46,4835546) after removing an additional 484 SNPs and 31,374 bp from the background. Only one SNP on DAARVY010000048.1 was reported as the vertically transmitted SNP. All three contigs from which SNPs were filtered match a known *Escherichia coli* plasmid (accession: CP023356.1), among others. Software for dTOURS and input and output files for regions filtered for this example are included in the data released at https://doi.org/10.5281/zenodo.14532372

## Simulations

One million sets were simulated for each of the thirty-five combinations of five small number of positions N chosen randomly from seven larger number of positions G. We chose N = {10, 25, 50, 75, 100} to cover the range for the number of vertical transmissions expected in close isolates in various species and G (in millions) = {1.5, 2.5, 3.5, 4.5, 5.5, 6.5, 12.7} for the size of bacterial and fungal pathogens. For each set S, we calculated the maximum RS found in the first iteration by dTOURS that indicates the RS value that is achieved at random by S.

For comparing computational resource requirements and sensitivity of dTOURS with Gubbins and FDA method, as the time needed by Gubbins for millions of pairs is large, we randomly subsampled 25,000 out of one million sets for each combination.

## Published outbreaks analyzed

To assess the quality of dTOURS and to show the importance of removing dense regions for detecting close isolates, we chose one published outbreak for each of the five pathogens found in foodborne outbreaks that are very common or cause high mortality rate. These are *Campylobacter jejuni*, *Legionella pneumophila*, *Listeria monocytogenes*, *Escherichia coli*, and *Salmonella Typhimurium*.

The *Campylobacter jejuni* outbreak was epidemiologically verified by CDC and included in the benchmark datasets [24]. This outbreak was caused in 2008 by contaminated raw milk consumption from a dairy farm in Pennsylvania, USA [30]. Publication showed that 22 isolates in the study were in four subclades of size 14, 4, 1, and 3.

The *Legionella pneumophila* outbreak [25] occurred in 2013 at Wesley hospital in Brisbane, Australia. The study had 46 isolates sequenced of which two were historical isolates and remaining 44 isolates were found to be distributed in three clonal subgroups of size 21, 9, and 14. The three clonal subgroups corresponded to geographically distinct sections of the hospital plumbing system.

The *Listeria monocytogenes* study [26] collected 252 isolates from four Norwegian meat processing plants between 2009 and 2017. Three isolates were found to be outliers while the remaining 249 isolates clustered in two clades of size 137 and 112. The study revealed cross-contamination between and within factories as well as persistent presence of contamination over several years.

The *Escherichia coli* investigation [27] in 2016 implicated raw flour as the source of infection in 56 cases spread over 24 states in USA. Whole genome shotgun sequence analysis of 54 isolates in the publication had 40 cases and 9 flour sample isolates as clonal while 5 isolates were outliers.

The *Salmonella Typhimurium* study [28] used two foodborne outbreaks along with sporadic cases between January and May 2014 in metropolitan Sydney, Australia. The study found only 30 SNPs in 62 isolates sequenced. It also found distinct patterns of SNPs for each of the two outbreaks.

## Data analysis

Sequence reads for all runs in all datasets analyzed were assembled using SKESA [31]. Assemblies for genomes submitted to NCBI without reads were downloaded from the NCBI assembly database. To compute SNPs for all pairs in a dataset, genomes in each pair were aligned using BLAST where the genome with higher N50 was used as the subject and the other genome as the query. SNPs for each pair were filtered using dTOURS for comparing them to the published results.

Gubbins concatenates contigs in the order contigs are given in the reference assembly to create a single sequence for the assembly. It also uses 17-mer based SKA analysis for SNP calling by default. We found cases where 17-mer mappings leading to SNP calls are not supported by read/contig alignment. An example is SNP calls on SKESA assemblies of SRR3657301 and SRR3747657 where the SKA alignment for a region with a SNP has a 32 bp alignment flanked by 47 and 60 gaps on 5' and 3' ends. The 32 bp sequence in the alignment from SRR3657301 (CGTCATACTCGACGCCGT-TAATCACCGCCTGT) has matches for the first and last 17 bases to two different contigs with no longer alignment. This illustrates that the 32 bp sequence is not supported by any contig in SRR3657301.

To reduce discrepancies due to contig ends and SNP calling methods when comparing dTOURS to Gubbins, for each isolate in every pair, we created a single sequence for each genome in a pair using regions that align between the pair. We used the SNPs called using BLAST alignments of these sequences for dTOURS and to introduce gaps in the sequences used as input to Gubbins. That is, the same set of SNPs were provided for filtering to both dTOURS and Gubbins in the format they use.

Comparison of Gubbins and dTOURS gives a set of regions filtered by one method but not the other. To check a region R reported only by Gubbins, we computed RS value for R using the background frequency in the result reported by dTOURS. For cases where a subsequence S was reported by dTOURS for a longer region R reported by Gubbins, the SNP count and length of S were added back to the background values in the result reported by dTOURS before computing RS for R. There were no cases where Gubbins and dTOURS reported overlapping regions where some part of the region was common to both and some exclusive to each method.

To test whether a region matches known plasmids or phages, we created databases for plasmid and phage sequences available in the Nucleotide database at NCBI on November 19, 2024 using query "plasmid[Filter] AND Bacteria [Organism]" and "phage[Title] OR bacteriophage[Title]", respectively. Regions tested were aligned to above databases using BLAST with word size 16, no low-complexity filtering, and default parameters for the rest.

To show that dTOURS is more robust than Gubbins for fragmented assemblies, we present an example where concatenating contig sequences in an order different than the one provided in the reference assembly produces a different result with Gubbins. Furthermore, analysis for two pairs in outbreaks Illustrates higher precision achieved by dTOURS.

## Results

### Parameter computation from simulation

In all 35 simulated combinations with one million sets each, the total number of sets that had RS above 25 was 330. The range was zero sets above RS of 25 in a few combinations to the maximum of 89 sets in the combination of 25 positions out of 6.5 million. The maximum RS for 25 out of 6.5 million combination was 25.02 while the maximum RS reached by any set was 31.02 for the combination of 50 positions out of 1.5 million. The distribution of maximum RS for a few combinations is shown in Fig 3. These show that, when looking for pairs of microbial genomes that are close, choice of 25 as the threshold is unlikely to remove regions that do not have a high density of variations.

The total CPU time for all million sets in a combination varied from 5 seconds to the maximum of 11 minutes 19 seconds for the combination of 100 positions chosen at random from 1.5 million. The runs were done on a shared compute farm, using various machines under different network loads. All runs were limited to a maximum of 2 GB of memory. This shows that running dTOURS is fast and requires minimal compute resources.

### Comparison on simulated sets

We compared dTOURS to Gubbins and the method used by FDA. As the FDA method only looks for 3 positions in a window of 1000 bases, it is faster than dTOURS. Time taken by Gubbins for 25,000 sets in the 35 combinations varied from 3 hr 42 minutes to 12 hr 57 minutes that is orders of magnitude more than both dTOURS and FDA method. Gubbins also writes temporary files that creates a significant disk management overhead for millions of runs.

Table 1 shows the number of sets out of 25,000 in each of the combinations reported to have a dense region. Total count is 8, 513, and 13512 for dTOURS, Gubbins, and FDA, respectively. For the three sets reported by dTOURS but not by FDA, the number of positions removed by dTOURS in each set is 2; below the count of 3 used by FDA. These results show that with the threshold of 25, dTOURS finds less random sets than both Gubbins and FDA method and is also computationally better suited than Gubbins for detecting dense regions in large number of close pairs.

### Analysis of *Campylobacter jejuni* outbreak

The *Campylobacter jejuni* outbreak had 22 isolates in four clades. The separation of isolates by published clades was reproduced in our analysis. S1 Table shows the number of SNPs for all pairs. The lower and upper triangles show SNP counts before and after filtering by dTOURS, respectively. It shows that while the number of SNPs within a clade are essentially not affected by density filtering, the distance between clades would be large without the filtering.

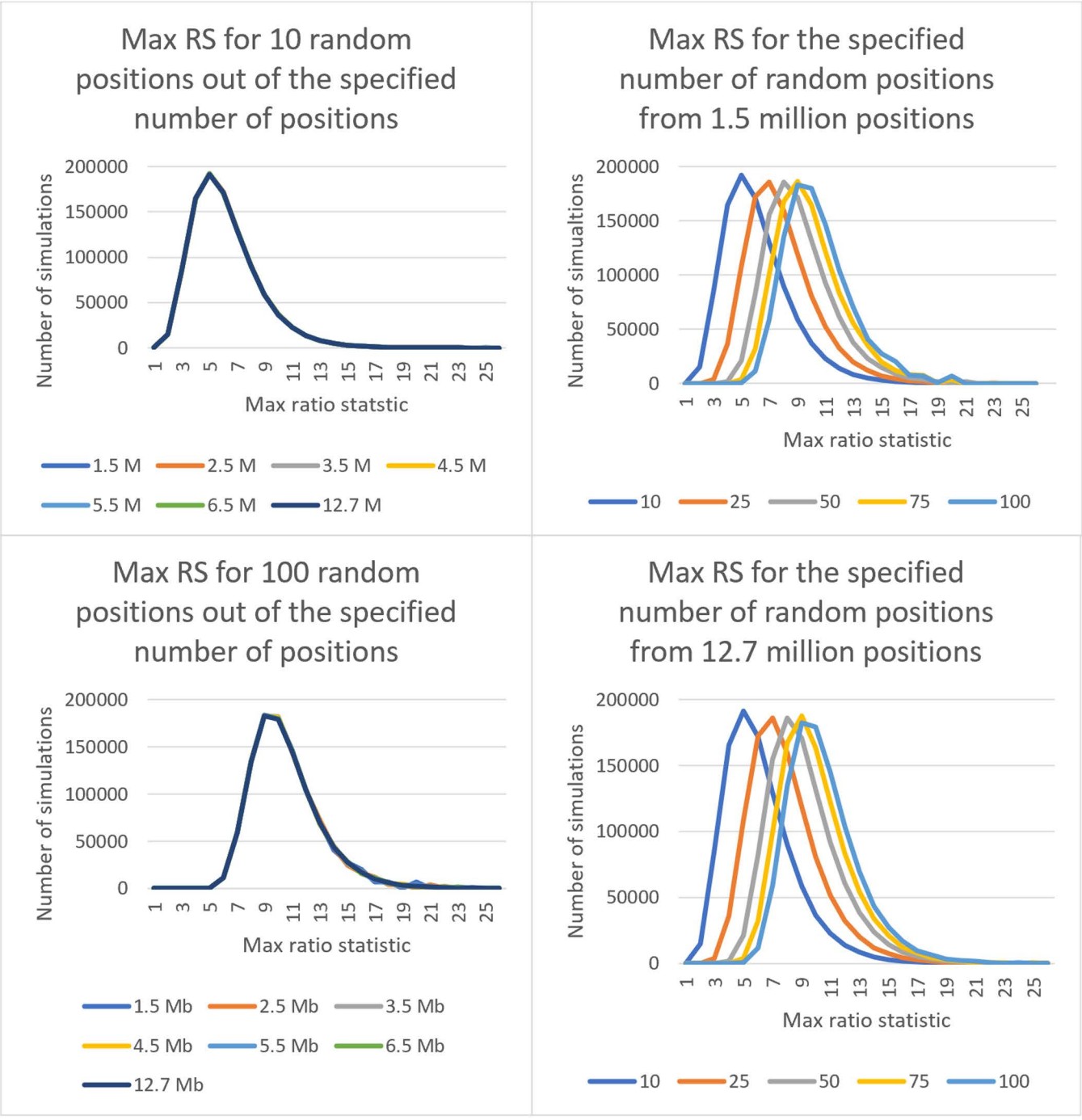

**Fig 3. Distribution of RS score for some simulated combinations.** With N={10, 25, 50, 75, 100} and G={1.5, 2.5, 3.5, 4.5, 5.5, 6.5, 12.7} million, combinations shown are (A) 10 positions from all counts in G (B) 100 positions from all counts in G (C) all positions in N from 1.5 million (D) all positions in N from 12.7 million.

Table 1. Number of sets with a dense region. Counts are out of 25,000 sets for each combination of specified number of positions randomly chosen from the larger number of positions. Rows for 10 positions out of 6.5 million and 12.7 million are not shown as no sets were reported by any method.

| #Positions | dTOURS | Gubbins | FDA | dTOURS | Gubbins | FDA | dTOURS | Gubbins | FDA |
|---|---|---|---|---|---|---|---|---|---|
| | 1.5 million | | | 2.5 million | | | 3.5 million | | |
| 10 | | 5 | 5 | | 2 | 2 | 1 | 2 | |
| 25 | | 12 | 81 | | 11 | 27 | | 4 | 11 |
| 50 | | 31 | 627 | | 16 | 220 | | 8 | 132 |
| 75 | | 44 | 2037 | | 34 | 809 | | 22 | 405 |
| 100 | | 50 | 4474 | 1 | 42 | 1773 | | 37 | 957 |
| | 4.5 million | | | 5.5 million | | | | | |
| 10 | 2 | 1 | | | 1 | | | | |
| 25 | | 5 | 6 | | 3 | 4 | | | |
| 50 | | 9 | 68 | | 6 | 43 | | | |
| 75 | | 18 | 244 | | 18 | 153 | | | |
| 100 | | 26 | 524 | | 21 | 369 | | | |
| | 6.5 million | | | 12.7 million | | | | | |
| 25 | 3 | 6 | 5 | 1 | 2 | 2 | | | |
| 50 | | 13 | 37 | | 6 | 10 | | | |
| 75 | | 20 | 121 | | 8 | 34 | | | |
| 100 | | 18 | 257 | | 12 | 75 | | | |

Ignoring the local insertion or deletions of less than 28 bases, our analysis shows that all regions removed correspond to thirteen homologous regions on the genomes where each region has two to seven distinct sequences among the 22 isolates. For each region, the mapping of different sequences seen for the region in each isolate is shown in Table 2. Isolates with the same sequence patterns in Table 2 provide additional support for the published clades. All 38 sequences for 13 regions encoded in Table 2 are available in S2 File with 'cj' as the prefix in sequence identifiers.

## Analysis of *Legionella pneumophila* outbreak

The *Legionella pneumophila* outbreak in Brisbane had 46 sequenced isolates. Two were historical isolates and remaining 44 were distributed in four clades in the publication. In our analysis, one historical isolate (LP42) had at least 77,944 SNPs with respect to the remaining 45 isolates before dTOURS filtering and at least 30,764 SNPs after filtering. Ignoring LP42, the other historical isolate (LP43) had SNP counts in range from 3,647 to 5,279 before filtering and 40 to 83 after filtering. S3 Table shows pairwise SNP counts before and after filtering for the remaining pairs of genomes from 44 isolates. It shows that filtered SNP counts support the published clade assignments. It also shows that without dTOURS filtering, ERR440097, ERR440100, ERR440082, and ERR440095 would not be assigned correctly to clades I, II, III, and III, respectively, and all three clades would appear much more distant from each other than their relatively recent divergence from a common ancestor.

Ignoring the local insertion or deletions of less than 28 bases, our analysis shows that all regions removed correspond to fourteen homologous regions on the genomes. One sequence for all 14 regions is available in S2 File with 'lp' as the prefix in sequence identifiers. Thirteen of these regions that match known plasmids have best matches to one of CP045733.1, FQ958212.1, or LN681226.1 as shown in S4 File. The sequence for the region with no matches to a known plasmid or phage is 'lp.region.04' that is 170 bp long. It has full length match to only 15/44 isolates and partial matches to ends of contigs for ten additional isolates, indicating a region that is not assembling properly and should not be used for SNP calling.

**Table 2. Regions removed and assignment of sequences seen in different genomes for each region.** Value (1,S) means that sequence is same as the one assigned value 1 but is found split on more than one contig. Value (2,3) means that the region is split into two sequences and the sequence is different from the rest. (N) means no alignments and (P) means alignments to only a part of the sequence. Each value in the table has a different color to make it easy to see patterns in the data.

| Clade | Set/Region | R01 | R02 | R03 | R04 | R05 | R06 | R07 | R08 | R09 | R10 | R11 | R12 | R13 |
|---|---|---|---|---|---|---|---|---|---|---|---|---|---|---|
| 1 | D7316 | 1 | 1 | 1 | 1 | 1 | 1 | 1 | 1 | 1 | 1 | 1 | N | N |
| | D7319 | 1 | 1 | 1 | 1 | 1 | 1 | 1 | 1 | 1 | 1 | 1 | 1 | N |
| | D7321 | 1 | 1 | 1 | 1 | 1 | 1 | 1 | 1 | 1 | 1 | 1 | N | N |
| | D7322 | 1 | 1 | 1 | 1 | 1 | 1 | 1 | 1 | 1 | 1 | 1 | 3 | N |
| | D7323 | 1 | 1 | 1 | 1 | 1 | 1 | 1 | 1 | 1 | 1 | 1 | N | 1 |
| | D7324 | 1 | 1 | 1 | 1 | 1 | 1 | 1 | 1 | 1 | 1 | 1 | N | N |
| | D7327 | 1 | 1 | 1 | 1 | 1 | 1 | 1 | 1 | 1 | 1 | 1 | N | N |
| | D7328 | 1 | 1 | 1 | 1 | 1 | 1 | 1 | 1 | 1 | 1 | 1 | N | N |
| | D7329 | 1 | 1 | 1 | 1 | 1 | 1 | 1 | 1 | 1 | 1 | 1 | N | N |
| | D7330 | 1 | 1 | 1 | 1 | 1 | 1 | 1 | 1 | 1 | 1 | 1 | N | N |
| | D7333 | 1 | 1 | 1 | 1 | 1 | 1 | 1 | 1 | 1 | 1 | 1 | 1 | N |
| | D7334 | 1 | 1 | 1 | 1 | 1 | 1 | 1 | 1 | 1 | 1 | 1 | 1 | N |
| | D7320 | 1 | 1 | 1 | 1 | 1 | 1 | 1 | 1 | 1 | 1 | 6 | N | N |
| | D7331 | 1 | 1 | 1 | 1 | 1 | 1 | 1 | 1 | 1 | 1 | 6 | N | N |
| 2 | 2014D-0067 | 2 | 1 | 1 | 1 | 3 | 3 | 2 | 2 | 2 | 3 | 3 | 1 | N |
| | 2014D-0070 | 2 | 1 | 1 | 1 | 3 | 3 | 2 | 2 | 2 | 3 | 3 | 1 | N |
| | 2014D-0189 | 2 | 1 | 1 | 1 | 3 | 3 | 2 | 2 | 2 | 3 | 3 | 1 | N |
| | 2014D-0068 | 2 | 1 | 1 | 1 | 3 | 3 | 2 | 2 | 2 | 3 | 7 | 1 | N |
| 3 | D5663 | 1 | 1 | 1 | 1 | 1 | 2 | 2 | 2 | 2 | 2 | 2 | 2 | N |
| 4 | PNUSA000195 | 1 | 2,3 | 2 | 2 | 2 | 1,S | 2 | 2 | 2 | 4 | 4 | 1,3,P | 2 |
| | PNUSA000194 | 1 | 2,3 | 2 | 2 | 2 | 1,S | 2 | 2 | 2 | 5 | 5 | 1,3,P | 2 |
| | PNUSA000196 | 1 | 2,3 | 2 | 2 | 2 | 1,S | 2 | 2 | 2 | 5 | 5 | 1,3,P | 2 |

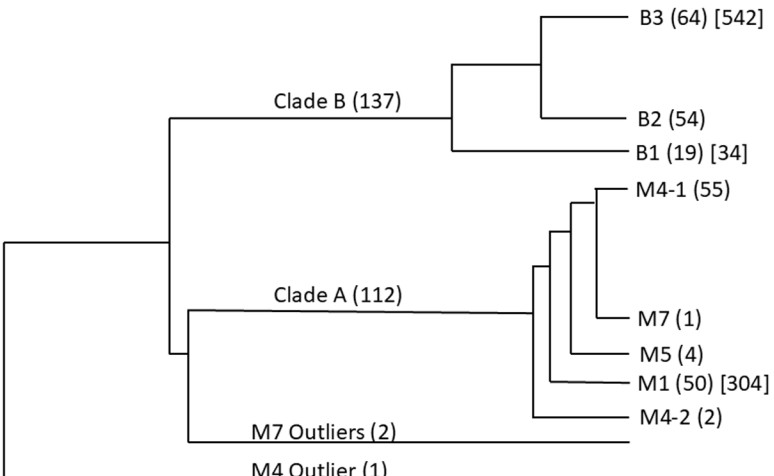

**Fig 4. Clades in Listeria outbreak.** Number of members in each clade and subclade are shown in round brackets. Number of pairs in each subclade where the unfiltered SNP count is at least 100 more than the filtered SNP count in our analysis is shown in square brackets.

## Analysis of *Listeria monocytogenes* outbreak

The *Listeria monocytogenes* outbreak had 252 isolates; three isolates were reported as outliers and remaining isolates were reported in two major clades A and B as shown in Fig 4 (clade structure from Fig 4 of [26]). In our analysis, single linkage clustering using SNP counts after dTOURS filtering retaining pairs with count at most 15 produces one group for all isolates in Clade A and divides the remaining sets correctly. Using SNP count at most 6 subdivides Clade A correctly as well, indicating that the structure and distance of clades and subclades is reproduced in the filtered SNP counts.

Out of 6,335 pairs within each of the seven subclades in Clades A and B, our analysis finds 880 pairs where the difference between unfiltered and filtered counts is at least 100. As shown in Fig 4, these 880 pairs are in Clade A branch M1 (304 pairs), Clade B branch B1 (34 pairs) and Clade B branch B3 (542 pairs). Regions removed by dTOURS from these 880 pairs correspond to 16 homologous regions. All 16 regions have alignments to known plasmids and phages for at least 75% of the sequence; eleven regions have alignments for over 92% of the sequence. One sequence for all 16 regions is available in S2 File with 'lm' as the prefix in sequence identifiers. Alignments for the sequences used for computing the statistics reported are available in S4 File.

## Analysis of *Escherichia coli* outbreak

Out of 54 isolates in the outbreak, as per the publication, all pairs from 49 isolates differ by only a couple of SNPs while 5 isolates are not part of the outbreak. Analysis of PNUSAE002540 (SRR3228456) shows that there is a data issue with reads available for this isolate in SRA as over 91% of the reads are for Salmonella. In our analysis, all 1128 pairs from the remaining 48 isolates in the outbreak have maximum SNP count of 12 after dTOURS filtering. There are 42 pairs where the unfiltered count is at least 100 more than the filtered count. Regions removed by dTOURS from these 42 pairs correspond to 25 homologous regions. All 25 regions have alignments to known plasmids and phages for over 97% of the sequence. One sequence for all 25 regions is available in S2 File with 'ec' as the prefix in sequence identifiers. Alignments for the sequences used for computing the statistics reported are available in S4 File.

## Analysis of *Salmonella Typhimurium* outbreak

All 1891 pairs from 62 isolates in the outbreak are very close to each other after dTOURS filtering. The maximum filtered count is 16. There are 12 pairs where the unfiltered count is at least 100 more than the filtered count; these 12 pairs have unfiltered count of over 200. Regions removed by dTOURS from these 12 pairs correspond to 19 homologous regions. All 19 regions have alignments to known plasmids and phages for over 97% of the sequence. One sequence for all 19 regions is available in S2 File with 'st' as the prefix in sequence identifiers. Alignments for the sequences used for computing the statistics reported are available in S4 File.

## Comparison to Gubbins

Both Gubbins and dTOURS use ratio statistics but have different implementations for how the windows for computing the statistic are created, processing done in each iteration, and how iterations are terminated. Gubbins uses a default value of four iterations while dTOURS uses an empirically determined RS value threshold of 25 for close isolates in bacterial and fungal species. Gubbins also uses fixed window sizes that is a reasonable approach when considering both diverged and close species. However, when looking for only close isolates, dTOURS can afford to be precise and look at all potential windows. Implementation in dTOURS also does not depend on requiring a single sequence for the genome whereas concatenating contigs in different orders to form a single sequence can lead to different results with Gubbins for the same input SNPs. An example of such an occurrence is presented in the next section.

Even with differences in implementation, when a single sequence and same input SNPs for comparison are created as described in the Methods section, both Gubbins and dTOURS give similar results. This is reflected in the $R^2$ value of

0.995 for the filtered SNP counts for 36,072 pairs in five outbreaks, excluding pairs with the two historical isolates in *Legionella pneumophila* study and PNUSAE002540 for *Salmonella Typhimurium* outbreak that has data issues. Per species, the $R^2$ values in outbreaks are 0.959 for 231 pairs of *Campylobacter jejuni*, 0.995 for 946 pairs of *Legionella pneumophila*, 0.994 for 31,626 pairs of *Listeria monocytogenes*, 0.999 for 1,378 pairs of *Escherichia coli*, and 0.956 for 1,891 pairs of *Salmonella Typhimurium*.

Among the 36,072 pairs, we investigated two pairs with biggest difference in counts of SNPs between the two methods where at least one method reported at most 50 SNPs. First pair below reports more SNPs with Gubbins than dTOURS while the second pair reports more SNPs with dTOURS than Gubbins. Sequences for the regions analyzed and their alignments are available in S2 and S4 Files, respectively, with 'pair' as the prefix in sequence identifiers.

**Pair 1:** For the pair SRR3476805 and SRR3476797, both methods remove over 500 SNPs. dTOURS retains only 3 SNPs while Gubbins retains 6 additional SNPs filtered by dTOURS. These 6 SNPs are from a 16,305 bp region with RS value of 28.46. Fig 5 shows these SNPs from contig 67 in SRR3476797 assembly along with two additional regions on the same contig where SNPs are removed by both methods. SNP counts shown are as reported by Gubbins; dTOURS reports 61 SNPs instead of 56 in the second region, including indels. The entire contig aligns to *Escherichia coli* plasmid LR890577.1 at >98.9% identity.

**Pair 2:** For the pair D5663 and S2014D-0070, input has 541 SNPs in 1,598,998 bp of aligned sequence where both methods remove 500 SNPs and retain 26 SNPs. Remaining 15 SNPs are filtered by Gubbins but retained by dTOURS. The length of regions filtered by dTOURS in 500 SNPs is 23,774 bp. Fifteen SNPs removed by Gubbins but not by dTOURS come from four different regions where none of the regions match a known plasmid or phage. These regions are as follows:

1. Gubbins removes 106 SNPs from a 7,102 bp region whereas dTOURS removes 103 SNPs from a 4,877 bp subsequence of the same region, leaving 3 SNPs at the 3' end of the region. RS value of 103 SNPs from 4,877 bp is 510.61 (from n,g,N,G = 103,4877,144,1580101) while RS for 106 SNPs from 7,102 bp is 490.78 (from n,g,N,G = 106,7102,144,1580101). Therefore, dTOURS chooses to remove 103 SNPs instead of 106.

2. Gubbins removes 5 SNPs from a 790 bp region that achieves RS value of 22.82 (from n,g,N,G = 5,790,41,1575224) that is below the threshold of 25.

3. Gubbins removes 3 SNPs from a 64 bp region that achieves RS value of 19.67 (from n,g,N,G = 3,64,41,1575224) that is below the threshold of 25.

4. Gubbins removes 4 SNPs from a 782 bp region that achieves RS value of 17.36 (from n,g,N,G = 4,782,41,1575224) that is below the threshold of 25.

Above analysis shows that for the first pair where dTOURS removes additional SNPs, region removed matches a known plasmid. For the second pair where dTOURS does not remove some SNPs, they are either extension of a region where dTOURS removes some SNPs or below the empirically determined RS threshold of 25. Also, these regions do not

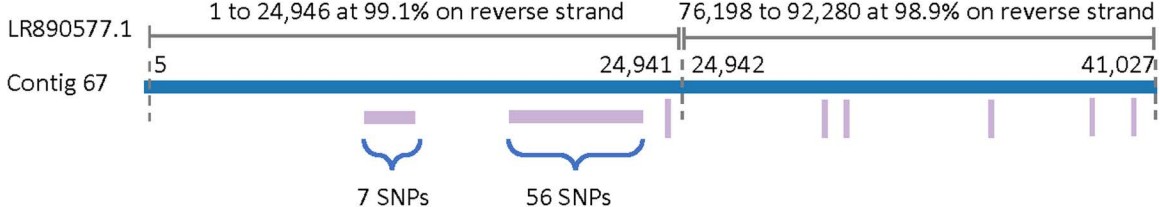

**Fig 5. Plasmid alignment and SNPs on contig 67 in SRR3476797 assembly with respect to SRR3476805 assembly.** Length of contig 67 in SRR3476797 assembly by SKESA and plasmid LR890577.1 is 41,027 bp and 92,280 bp, respectively. Each vertical pink bar shows a single SNP.

have matches to known plasmids or phages. These analyses show that dTOURS is more precise than Gubbins for detecting close isolates.

### Contig order affects Gubbins filtering

We illustrate that same input assemblies can give different results using Gubbins if the contigs are ordered differently. Illustration uses SNPs from an outbreak pair SRR3294546 and SRR3657299 where we noticed a difference between dTOURS and Gubbins filtering. The relevant SNPs in the SKESA assembly of SRR3294546 are at positions 1989 and 1997 of contig 6 and positions 15897, 16096, 16097, 16100, 16261, and 16263 of contig 7 where the length of contig 6 and contig 7 is 2,031 bp and 18,230 bp, respectively. When the contigs are concatenated in the same order as in assembly for SRR3294546, Gubbins does not filter two SNPs on contig 6. However, if contig 6 is reverse completed and concatenated after contig 7, then all eight SNPs are filtered by Gubbins.

Concatenated sequences in FASTA format for the original order and reordering as specified above are available in S5 and S6 Files, respectively. The first sequence in both files is assembly for SRR3294546 while the second sequence in both files introduces only the eight SNPs mentioned above for illustration purpose.

### Discussion

Explicitly identifying regions dense in point differences as done with dTOURS is useful for analysis of ensembles of bacterial isolates. Explicitly enumerated horizontally transferred regions are often retrospectively identifiable as plasmids; we can identify the extent of these without dependence on a preexisting set of known plasmids formatted for analysis software. Our approach here is also free of any dependency on availability of preexisting reference genomes with the core regions enumerated, that is, free of dependency on a reference which has a partial recombination-free region annotated. Our approach of explicitly enumerating the filtered region in pairwise comparison also allows identification of which isolates are most strongly affected by the presence of horizontally transferred regions. The core of the algorithm is a log-odds calculation, so scoring is necessary to identify a meaningful reporting threshold. We present simulations that support the choice of reporting threshold that we have used.

This standalone implementation of a density filtering algorithm has been applied for analysis of bacterial isolates generated during public health monitoring. Researchers interested in the possibility of creating similar pipelines with other modular components (alignment, assembly, variation calling), motivated perhaps by evolving sequencing technology, need an algorithm like the one implemented in dTOURS for identifying horizontally transferred regions.

dTOURS has potential to be used in other contexts for identifying the existence and boundaries of clusters of discrete events, though the only application we have explored to date is point differences among bacterial isolates. For example, the distribution of RNA editing sites in Alu and LINE elements is known to have roughly lognormally variable local density [32]. Rough estimation using ad-hoc density thresholds [32,33] is a common approach to identifying and characterizing these clusters. Computational implementation of a probabilistic basis in dTOURS might be useful for analyses of these events in Alu or LINE elements.

### Supporting information

**S1 Table. SNP counts for Campylobacter dataset.** Lower triangle and upper triangle show SNP counts before and after SNP dense region removal by dTOURS, respectively. The cells in the table are color coded using 2-color scale from green to red where small values have cells colored green and cells with large values have red color.
(XLSX)

**S2 File. Sequences for regions analyzed in five outbreaks and for comparison to Gubbins.** A total of 117 sequences in FASTA format present in this file contain 38, 14, 16, 25, and 19 sequences from *Campylobacter jejuni,*

*Legionella pneumophila, Listeria monocytogenes, Escherichia coli,* and *Salmonella Typhimurium* outbreak analysis, respectively, and remaining five from two pairs in comparison to Gubbins.
(GZ)

**S3 Table. SNP counts for Legionella dataset.** Lower triangle and upper triangle show SNP counts before and after region removal by dTOURS, respectively. The cells in the table are color coded using 2-color scale from green to red where small values have cells colored green and cells with large values have red color.
(XLSX)

**S4 File. Alignments to plasmid and phage databases.** Alignments are reported in BLAST tabular format with columns for query, subject, percent identity, alignment length, number of mismatches, number of gaps, query start, query end, subject start, and subject end. If the alignment is on reverse strand, then subject start is greater than subject end.
(TXT)

**S5 File. Contigs concatenated in same order as in the assembly produced by SKESA.** This file contains two sequences. The first sequence has all contigs concatenated in the same order as in the assembly produced by SKESA for SRR3294546. The second sequence introduces eight SNPs in the first sequence.
(GZ)

**S6 File. Contigs concatenated with one contig moved and reverse complemented.** Two sequences in this file correspond to the two sequences in S5 except one contig is reverse complemented and ordered differently in the concatenation.
(GZ)

## Acknowledgments

We would like to thank David Lipman and the Pathogen detection pipeline team for useful discussions and for providing feedback on the manuscript.

## Author contributions

**Conceptualization:** Richa Agarwala.

**Formal analysis:** Richa Agarwala.

**Methodology:** Lukas Wagner.

**Software:** Richa Agarwala.

**Validation:** Richa Agarwala.

**Writing – original draft:** Richa Agarwala.

**Writing – review & editing:** Lukas Wagner, Richa Agarwala.

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
