## [Decision Letter · Decision Letter 0]

2 Feb 2025

PONE-D-24-59061dTOURS: dense-region tagging for outbreak detection using ratio statisticsPLOS ONE

Dear Dr. Agarwala,

Thank you for submitting your manuscript to PLOS ONE. After careful consideration, we feel that it has merit but does not fully meet PLOS ONE’s publication criteria as it currently stands. Therefore, we invite you to submit a revised version of the manuscript that addresses the points raised during the review process.

We look forward to receiving your revised manuscript.

Kind regards,

Rahul Shubhra Mandal, Ph.D.

Academic Editor

PLOS ONE

Journal Requirements:

“This research work was supported by the National Center for Biotechnology Information of the National Library of Medicine (NLM), National Institutes of Health. The contents of this publication are solely the responsibility of the authors and do not necessarily represent the official views of the National Center of Biotechnology Information.” 

“This research work was supported by the National Center for Biotechnology Information of the National Library of Medicine (NLM), National Institutes of Health. The contents of this publication are solely the responsibility of the authors and do not necessarily represent the official views of the National Center of Biotechnology Information. We would like to thank David Lipman and the Pathogen detection pipeline team for useful discussions and for providing feedback on the manuscript. “

‘This research work was supported by the National Center for Biotechnology Information of the National Library of Medicine (NLM), National Institutes of Health. The contents of this publication are solely the responsibility of the authors and do not necessarily represent the official views of the National Center of Biotechnology Information.”

Reviewers' comments:

Reviewer's Responses to Questions

**Comments to the Author**

1. Is the manuscript technically sound, and do the data support the conclusions?

Reviewer #1: Yes

Reviewer #2: Yes

2. Has the statistical analysis been performed appropriately and rigorously? 

Reviewer #1: Yes

Reviewer #2: Yes

3. Have the authors made all data underlying the findings in their manuscript fully available?

Reviewer #1: Yes

Reviewer #2: Yes

4. Is the manuscript presented in an intelligible fashion and written in standard English?

Reviewer #1: Yes

Reviewer #2: Yes

5. Review Comments to the Author

Reviewer #1: This paper introduces dTOURS, a ratio statistic-based model for identifying higher-density SNPs in whole genomes. dTOURS demonstrates improved performance over the current state-of-the-art (SOTA) model, Gubbins, and is more versatile and robust. The case study of five pathogen outbreaks highlights the competitiveness of this model. I recommend acceptance of the paper, provided the following points are addressed:

1. Include a summary figure of the dTOURS algorithm to give an overview of the approach and enhance the paper's readability.

2. Provide results for Gubbins on simulated data for comparison. If not already conducted, please include these results.

3. Clarify the color coding used in Tables 1–3.

4. Ensure consistent font sizes across Table 1 and Table 2 (e.g., "Clade 3" appears unequal).

5. While the paper focuses on pathogen outbreaks, like Gubbins, dTOURS likely has broader applications. It will benefit the community if the discussion includes potential additional applications of dTOURS.

Reviewer #2: Dear Authors,

Your manuscript presents a valuable contribution to pathogen outbreak genomics with the development of dTOURS. To further strengthen its clarity and impact, I recommend refining the explanation of the ratio statistic, incorporating a computational efficiency analysis, and expanding comparisons with additional SNP detection tools beyond Gubbins. Enhancing result visualization with annotated figures and addressing potential false positives and negatives will further improve the manuscript’s rigor and reproducibility. These revisions will ensure that dTOURS is effectively positioned as a robust and widely applicable tool for outbreak detection. 

6. PLOS authors have the option to publish the peer review history of their article (what does this mean? ). If published, this will include your full peer review and any attached files.

**Do you want your identity to be public for this peer review?** For information about this choice, including consent withdrawal, please see our Privacy Policy .

Reviewer #1: No

Reviewer #2: No

---

## [Author Response · Author response to Decision Letter 1]

7 Mar 2025

Response letter has been uploaded

---

## [Decision Letter · Decision Letter 1]

25 Mar 2025

dTOURS: dense-region tagging for outbreak detection using ratio statistics

PONE-D-24-59061R1

Dear Dr. Agarwala,

We’re pleased to inform you that your manuscript has been judged scientifically suitable for publication and will be formally accepted for publication once it meets all outstanding technical requirements.

Kind regards,

Rahul Shubhra Mandal, Ph.D.

Academic Editor

PLOS ONE

Additional Editor Comments (optional):

Reviewers' comments:

Reviewer's Responses to Questions

**Comments to the Author**

1. If the authors have adequately addressed your comments raised in a previous round of review and you feel that this manuscript is now acceptable for publication, you may indicate that here to bypass the “Comments to the Author” section, enter your conflict of interest statement in the “Confidential to Editor” section, and submit your "Accept" recommendation.

Reviewer #1: All comments have been addressed

Reviewer #2: All comments have been addressed

2. Is the manuscript technically sound, and do the data support the conclusions?

Reviewer #1: Yes

Reviewer #2: Yes

3. Has the statistical analysis been performed appropriately and rigorously? 

Reviewer #1: Yes

Reviewer #2: Yes

4. Have the authors made all data underlying the findings in their manuscript fully available?

Reviewer #1: Yes

Reviewer #2: Yes

5. Is the manuscript presented in an intelligible fashion and written in standard English?

Reviewer #1: Yes

Reviewer #2: Yes

6. Review Comments to the Author

Reviewer #1: (No Response)

Reviewer #2: The revised manuscript is significantly better. I would recommend the addition of visualization tools like tables for outlining the distinctions in dTOURS and other tools, as well as sumamrising results for better readablility.

7. PLOS authors have the option to publish the peer review history of their article (what does this mean? ). If published, this will include your full peer review and any attached files.

**Do you want your identity to be public for this peer review?** For information about this choice, including consent withdrawal, please see our Privacy Policy .

Reviewer #1: No

Reviewer #2: **Yes: ** Priyanka Rawat

---

## [Editor Report · Acceptance letter]

PONE-D-24-59061R1

PLOS ONE

Dear Dr. Agarwala,

I'm pleased to inform you that your manuscript has been deemed suitable for publication in PLOS ONE. Congratulations! Your manuscript is now being handed over to our production team.

Kind regards,

on behalf of

Dr. Rahul Shubhra Mandal

Academic Editor

PLOS ONE